# Capacity and quality of maternal and child health services delivery at the subnational primary healthcare level in relation to intermediate health outputs: a cross-sectional study of 12 low-income and middle-income countries

Marwa Ramadan  , Jose Carlos Gutierrez  , Cameron Feil  , Sarah Bolongaita  , Oscar Bernal  , Manuela Villar Uribe 

Health, Nutrition and Population, The World Bank Group, Washington, DC, USA

**Correspondence to**
Dr Marwa Ramadan; marwa.eldesoky1987@gmail.com

## ABSTRACT

**Objectives** To examine the capacity and quality of maternal and child health (MCH) services at the subnational primary healthcare (PHC) level in 12 low-income and middle-income countries (LMICs) and its association with intermediate health outputs such as coverage and access to care.

**Design** Observational cross-sectional study using matched subnational data from service provision assessment surveys and demographic health surveys from 2007 to 2019.

**Settings** 138 subnational areas with available survey data in 12 LMICs (Afghanistan, Bangladesh, Democratic Republic of Congo, Haiti, Kenya, Malawi, Namibia, Nepal, Rwanda, Senegal, Tanzania and Uganda).

**Outcomes** Eight intermediate MCH outcomes/outputs were explored: (1) met need for family planning by modern methods; (2) attendance of four or more antenatal care visits; (3) perceived financial barriers to care; (4) perceived geographical barriers to care; (5) diphtheria-pertussis-tetanus (DPT) third dose coverage; (6) DPT dropout-rate; (7) care-seeking for pneumonia; and (8) oral rehydration solutions coverage.

**Results** Overall, moderate-to-poor PHC performance was observed across the 12 countries, with substantial heterogeneity between the different subnational areas in the same country as well as within the same subnational area across both capacity and quality subdomains. The analysis of the relationship between PHC service delivery and child health outcomes revealed that recent supervision (b=0.34, p<0.01) and supervisors' feedback (b=0.28, p<0.05) were each associated with increased care-seeking for pneumonia. We also observed the associations of several measures of capacity and quality with DPT immunisation. The analysis of maternal health outcomes yielded only a few statistically significant results at p<0.05 level, however, none remained significant after adjusting for other covariates.

**Conclusion** The results of this analysis illustrate the heterogeneity in the capacity and quality of PHC service delivery within LMICs. Countries seeking to strengthen their PHC systems could improve PHC monitoring at the subnational level to better understand subnational bottlenecks in service delivery.

## STRENGTHS AND LIMITATIONS OF THIS STUDY

⇒ Rather than focusing on primary healthcare (PHC) delivery at the national level, this study bridges a research gap by examining PHC service delivery at the subnational level and its association with improved health outputs in 12 low-income and middle-income countries.

⇒ The methodology used in the present study demonstrates how publicly available survey data such as the Demographic Health Survey (DHS) and the Service Provision Assessment (SPA) survey can be valuable tools in highlighting bottlenecks and disparities in PHC service delivery at the subnational level.

⇒ As the study is observational and cross-sectional, the association presented should not be interpreted as causal given the potential for an ecological bias.

⇒ Although the sample was constructed such that the time between the SPA and DHS surveys in each country was less than 5 years, it is possible that changes in time-varying factors could have influenced the outcomes under study.

⇒ There is a potential for omitted variable bias through other factors that may influence the capacity, quality and performance variables under study but are unmeasured in the SPA and DHS surveys.

## INTRODUCTION

Despite renewed global commitment to primary healthcare (PHC) in recent years, the measurement of PHC capacity and quality remains a challenge for policymakers seeking to improve the performance of health services. The transition from the

Millennium Development Goals to Sustainable Development Goals (SDGs) in 2016 marked a significant shift in the global health agenda. SDG 3, 'Achieving Universal Healthcare Coverage,' was a call to action to strengthen health systems, shifting the global health agenda away from vertically planned health programmes and initiatives. In 2018, 40 years after the Alma-Alta Declaration which identified PHC as the pathway to achieving health for all, global actors, governments and organisations reaffirmed a global commitment to PHC with the issuance of the Astana Declaration.[1]

More significant efforts are required to improve PHC performance in low-income and middle-income countries (LMICs).[2] Performance measurement has the potential to catalyse PHC improvement by identifying bottlenecks.[3 4] The Primary Health Care Performance Initiative (PHCPI), a global collaborative dedicated to measuring and improving PHC systems in LMICs, developed a conceptual framework illustrating the relationships between financing, inputs and core functions of the PHC system.[2] In 2018, PHCPI operationalised the conceptual framework into the Vital Signs Profile (VSP), which uses qualitative and quantitative indicators to provide a comprehensive understanding of PHC performance in a county.[3]

The relationship between capacity, quality and health system performance has long been studied. Donabedian captured the causal chain of inputs to outputs, commonly defined as the 'input-process-output-model', in 1966. More recent efforts by Randhawa[4] and Veillard et al[5] have demonstrated the relationship between inputs and outcomes of the PHC system. Previous studies also highlighted that PHC capacity, including the availability of providers, governance and financing, has been found to impact PHC-related outcomes in LMICs at the national level.[6–10] In addition, several studies have identified relationships between elements of high-quality PHC and performance at the national level.[2 11–14] However, the majority of associations were explored at the national level and significant gaps still exist in understanding PHC service delivery at the subnational level and its association with improved health outputs.

This paper aims to fill some of the current measurement gaps by exploring the relationship between PHC system capacity, quality and performance at the subnational level using quantitative indicators from the PHCPI VSP. Specifically, we assess PHC service delivery at both the national and subnational levels in 12 LMICs. In addition, we examine the association of PHC capacity and quality with intermediate health outputs such as coverage and access to services at the subnational level.

## METHODS
### Data sources
In this cross-sectional analysis, Service Provision Assessment (SPA) surveys and Demographic Health Surveys (DHS) were the main sources of data on PHC service delivery and PHC intermediate population health outputs, respectively, in 12 LMICs (Afghanistan, Bangladesh, DRC (Democratic Republic of Congo), Haiti, Kenya, Malawi, Namibia, Nepal, Rwanda, Senegal, Tanzania and Uganda). Specifically, we included all LMICs (as defined by the World Bank income classification) where both a DHS and an SPA survey were conducted within a 5-year interval in the period between 2007 and 2019. SPA surveys are comprehensive facility assessments that aim to collect information on the availability of services at the facility level, the readiness of facilities to provide these services, the compliance with acceptable standards of care and whether clients and service providers are satisfied with the service delivery process.[15] SPA surveys usually involve random selection of 400–700 facilities from a comprehensive sampling list of all public and private health facilities in a country or alternatively a complete census of facilities in the country. Facilities are usually categorised by facility type, managing authority (public and non-public), as well as subnational region; sampling and sample size ensure representativeness across each of these categories. Weights are applied during the analysis of these surveys to ensure that the contribution of selected facilities is proportional to their distribution in the country. On the other hand, DHS are nationally representative household surveys and a vital source of information on population, health and nutrition indicators in more than 90 countries. The standard DHS survey is conducted every 5 years using a large sample size (5000–30 000 households per survey), and usually uses a two-stage sampling methodology based on the availability of census information.[16] The DHS sampling and sample size ensure representativeness at the subnational region level. Table 1 provides additional information on the sampling structure and the time frame for data collection in each of the studied countries. The SPA and DHS data sets used in this study are publicly available and can be downloaded from the DHS website.[17]

### Metrics
Guided by the PHCPI framework and survey data availability, the capacity of the PHC system was assessed using the following five indicators: (1) availability of basic facility infrastructure; (2) the presence of recent supervision within the past 6 months; (3) the provision of feedback by supervisors; (4) the provision of outreach services; and (5) absence of PHC fees defined as free immunisation, Reproductive, Maternal, Newborn, and Child Health (RMNCH), non-communicable diseases and HIV services. Similarly, whenever data was available, PHCPI's VSP guided the assessment of PHC quality of care through the following six indicators: (1) waiting time of less than 60 min; (2) availability of providers defined as an average consultation time of more than 10 min; (3) competence of providers defined as the average of antenatal care, family planning and sick child quality scores based on WHO guidelines; (4) person-centredness defined as the per cent of clients who were told their diagnosis;

**Table 1** Characteristics of DHS and SPA surveys included in the analysis

| Country | Subnational areas | DHS field work | DHS sampling | DHS sample size (ever-married women) | Year data collection—SPA | SPA sample size (facilities) |
|---|---|---|---|---|---|---|
| Afghanistan | 34 provinces | June 2015 to February 2016 | Two-stage stratified sample | 24 395 | November 2018 to January 2019 | 160 facilities* |
| Bangladesh | 8 divisions | 24 October 2017 to March 2018 | Two-stage stratified sample | 20 100 | July 2017 to October 2017 | 1600 |
| DRC | 26 new provinces | August 2013 to February 2014 | Three-stage stratified sample† | 18 827 | October 2017 to April 2018 | 1380 |
| Haiti | 10 departments | November 2016 to April 2017 | Two-stage stratified sample | 14 371 | December 2017 to May 2018 | 1007 |
| Kenya | 8 regions | May 2014 to October 2014 | Two-stage stratified sample | 31 079 | January 2010 to May 2010 | 695 |
| Malawi | 3 regions | October 2015 to February 2016 | Two-stage stratified sample | 24 562 | June 2013 to February 2014 | 1060 |
| Namibia | 13 regions | May 2013 to September 2013 | Two-stage stratified sample | 9176 | July 2009 to October 2009 | 446 |
| Nepal | 7 provinces | June 2016 to January 2017 | Stratified sample (two stages in rural areas and three stages in urban areas) | 12 862 | April 2015 to November 2015 | 1000 |
| Rwanda | 5 provinces | September 2010 to March 2011 | Two-stage stratified sample | 13 671 | June 2007 to August 2007 | 538 |
| Senegal | 14 regions | April 2017 to December 2017 | Two-stage stratified sample | 16 787 | March 2017 to December 2017 | 3764 |
| Tanzania | 25 regions | August 2015 to February 2016 | Two-stage stratified sample | 13 266 | October 2014 to March 2015 | 1200 |
| Uganda | 15 regions | June 2011 to December 2011 | Two-stage stratified sample | 8674 | July to October 2007 | 491 |

*Sampling limited to urban areas of seven major provinces (Kabul, Nangarhar, Paktya, Kunduz, Balkh, Kandahar and Herat).
†Two-stage stratified sample in statutory towns and cities of established provinces, three-stage stratified sample in rest of established provinces and new provinces.
DHS, Demographic Health Surveys; DRC, Democratic Republic of Congo; SPA, Service Provision Assessment.

(5) safety defined as the presence of adequate infection control practices and waste disposal at the facility; and (6) comprehensiveness of PHC services defined as the average availability of tracer items for RMNCH, infectious diseases and non-communicable diseases. Additional information on PHCPI's framework and methodology is provided in an earlier publication.[5]

Additionally, eight intermediate maternal and child health outcomes/outputs were explored in accordance with the PHCPI VSP and consistent reporting of DHS indicators in the studied countries. Specifically, we assessed the following maternal and child health outputs: (1) met need for family planning by modern methods; (2) attendance of four or more antenatal care visits; (3) perceived financial barriers to care; (4) perceived geographical barriers to care; (5) diphtheria-pertussis-tetanus third dose (DPT3) coverage; (6) DPT dropout-rate; (7) care-seeking for pneumonia; and (8) oral rehydration solutions coverage. Other PHC-related outputs (eg, non-communicable diseases, tuberculosis and HIV) were not assessed due to the limited availability of subnational data in the studied countries. Detailed definition of each of the capacity, quality and output indicators are provided in online supplemental table 1.

### Data analysis
Population and health facility data were linked at the first administrative level (eg, region, province) whenever data was available and the gap between a population-based survey and a health facility survey was less than 5 years. In situations where substantial variation existed at the first administrative level between the two types of surveys, geographical coordinates of health facilities and DHS clusters were input into geographical information system software (ArcGIS) for matching. Specifically, in the DRC, DHS data in 2013 were reported for 11 formerly defined

provinces, while 2017 SPA data were reported for 26 newly defined provinces. ArcGIS was used to match subnational areas in both types of surveys. In addition, in Afghanistan, the SPA survey was only conducted in seven urban subnational areas, so the match process with DHS was limited to these areas (see online supplemental figure 1).

To compute subnational estimates, we applied the recommended SPA weights at the facility, provider and visit levels in the 12 studied countries. We then computed national averages based on the assumption that all subnational areas were equally weighted in each of the studied countries. In addition, we examined the subnational association between the computed service delivery subdomains and DHS intermediate maternal and child health outputs using bivariate and multivariable models with a significance level at p<0.05. Given the hierarchical nature of the examined population and health facility data, we used a two-level fixed-effects model to account for possible clustering of the observations at the country level. We also adjusted for the following population and facility characteristics at the subnational level: proportion of urban households, proportion of population in the fifth economic quintile (highest) and proportion of women with secondary or more education, proportion of private facilities (per the total number of health facilities at the subnational level), proportion of hospitals (per the total number of health facilities at the subnational level). The statistical analyses were all performed in Stata (V.17.0) and tables were created using Microsoft Excel.

**Patient and public involvement**
None.

**RESULTS**
**PHC service delivery at the national and subnational levels**
In the present study, the capacity and quality of PHC service delivery were analysed in 138 subnational areas in 12 LMICs. Substantial heterogeneity was observed across countries in the capacity subdomain. For example, the average availability of facility infrastructure (basic amenities and infection prevention) was as high as 80% in Afghanistan and as low as 48% in DRC. Most facilities in the analysed subnational areas reported the presence of recent supervision, with an average ranging from 94% in Bangladesh to 70% in Tanzania, while supervisor's feedback was provided in a relatively fewer number of facilities, with the highest average reported in Bangladesh (91%) and the lowest in Afghanistan (57%). Meanwhile, the provision of facility outreach services varied largely across the studied countries, with 84% of facilities in Nepal providing outreach services versus only 14% of facilities in the analysed subnational areas in Afghanistan. Similarly, large variation in charging PHC fees was observed across the sample; 99% of facilities in the analysed subnational areas of Uganda did not charge fees for PHC services versus only 32% in Tanzania. Table 2 shows the average score of individual capacity subdomains in the analysed subnational areas across the 12 studied countries.

**Table 2** The capacity and quality of PHC service delivery in the studied countries (average scores of subnational areas)

| Domain | Subdomain | Country | | | | | | | | | | | |
| --- | --- | --- | --- | --- | --- | --- | --- | --- | --- | --- | --- | --- | --- |
| | | AF18* | BD17 | DRC18 | HT17 | KE10 | MW13 | NM09 | NP15 | RW07 | SN17 | TZ15 | UG07 |
| Capacity | Facility infrastructure | 80% | 51% | 48% | 62% | 66% | 67% | 69% | 53% | 63% | 62% | 56% | 55% |
| | Recent supervision | 77% | 94% | 77% | 74% | 74% | 74% | 72% | 72% | 82% | 77% | 70% | 81% |
| | Supervisor feedback | 57% | 91% | 72% | 64% | 63% | 65% | 61% | 64% | 74% | 76% | 65% | 62% |
| | Outreach services | 14% | 18% | 73% | 39% | 43% | 70% | 20% | 84% | 79% | 48% | 50% | 79% |
| | Not charging PHC fees | 50% | 67% | 42% | 73% | 67% | 74% | 94% | 92% | 96% | 65% | 32% | 99% |
| | **Capacity score** | 55% | 65% | 61% | 60% | 61% | 70% | 63% | 73% | 79% | 64% | 53% | 76% |
| Quality | Waiting time <60 min | 92% | – | 76% | 58% | 61% | 48% | 44% | 94% | 37% | 77% | 57% | 49% |
| | Provider availability | 38% | – | 78% | 65% | 63% | 41% | 63% | 44% | 39% | 62% | 70% | 42% |
| | Provider competence | 30% | – | 39% | 40% | 41% | 37% | 54% | 31% | 41% | 40% | 40% | 49% |
| | Person centredness | 45% | – | 31% | 25% | 38% | 45% | 54% | 33% | 26% | 9% | 46% | 54% |
| | Safety | 31% | 39% | 32% | 55% | 65% | 49% | 73% | 45% | 26% | 65% | 41% | 26% |
| | Comprehensiveness | 68% | 63% | 74% | 79% | 81% | 75% | 85% | 76% | 75% | 65% | 71% | 72% |
| | **Quality** | 50% | 51% | 55% | 56% | 61% | 49% | 62% | 54% | 41% | 53% | 54% | 47% |

AF18: Afghanistan 2018, BD17: Bangladesh 2017, DRC18: Democratic Republic of Congo 2018, HT17: Haiti 2017, KE10: Kenya 2010, MW13: Malawi 2013, NM09: Namibia 2009, NP15: Nepal 2015, RW07: Rwanda 2007, SN17: Senegal 2017, TZ15: Tanzania 2015, UG07: Uganda 2007.
*Estimates represent the average of urban subnational data for which Service Provision Assessment and Demographic Health Surveys data were available.
PHC, primary healthcare.

When the quality-of-service delivery was assessed, most countries had moderate-to-poor scores in all quality subdomains except waiting time and comprehensiveness, where some countries performed well. As with capacity, cross-country variation in quality subdomains was observed. For example, the average waiting time was less than 60 min in more than 90% of the facilities in the analysed subnational areas of Afghanistan and Nepal versus only 37% of the analysed facilities in Rwanda. However, the average consultation time was more than 10 min in only 38% of the analysed Afghan facilities. Competence of providers was one of the quality subdomains with the lowest average scores across the studied countries ranging from 30% on average, for the analysed facilities in Afghanistan to 54% in Namibia. Similarly, person centredness scored poorly in most countries with an average score as low as 9% for the analysed facilities in Senegal versus 54% in Uganda. Meanwhile, at least two-thirds of the analysed facilities offered comprehensive PHC services including RMNCH, non-communicable and infectious disease services in all the studied countries (table 2).

Subnational analysis of service delivery subdomains revealed the presence of geographical disparities at the first administrative level in both PHC capacity and quality subdomains. For example, large subnational variation was observed in the average availability of facility infrastructure in Senegal, ranging from 83% in Diourbel region to 45% in Kedougou region. Similarly, considerable subnational variation was observed in the provision of outreach services in Haiti, ranging from 45% in the Northeast to 5% in the Northwest department. Charging of PHC fees in DRC was also another example of administrative variation in PHC capacity, where 81% of facilities in Sankuru province provided free PHC services versus none of the facilities in North-Kivu or South-Kivu provinces. Similarly, subnational variation was observed in most of the examined quality subdomains. For example, the range of variation across the different subnational areas in Tanzania was more than 60% points for waiting time. In addition, the range of variation was more than 50% for availability of providers across the different subnational areas in Senegal. Furthermore, person-centred care ranged from 89% in Kunduz province to only 1% in Kandahar province in Afghanistan. Meanwhile, the lowest subnational variations were observed in provider competence subdomain, where the majority of subnational areas scored below 50% in the 12 studied countries.

Not only did we observe variations within the same country across different subnational areas, but also within the same subnational area across both quality and capacity subdomains. For example, the range of variation across different capacity subdomains was more than 50% points in the majority of the analysed subnational areas in Afghanistan, Bangladesh and Namibia. Most of these variations were derived by the difference in the capacity of the facilities in providing outreach services and offering free PHC services. Similarly, several subnational areas in Afghanistan, DRC and Senegal showed substantial variation across the different quality subdomains. For example, the range of variation across quality subdomains in Kandahar, Afghanistan was 99%, where all facilities had a waiting time of less than 60 min; however only 1% provided person-centred care. Online supplemental table 2 and online supplemental table 3 provide detailed information on the capacity and quality scores in each of the analysed 138 subnational areas (see online supplemental material).

## The association between PHC service delivery subdomains and intermediate health outputs

Examining the relationship between subnational PHC capacity and maternal health outputs, no statistically significant association was found between the capacity of facilities and antenatal care coverage after adjustment for subnational characteristics. Meanwhile, a statistically significant association was observed between the general availability of basic facility infrastructure and perceived economic and geographical barriers to care. Specifically, a 10%-point increase in the average availability of basic infrastructure at the subnational level was associated with 3% lower perceived financial barriers to care (b=−0.35, p<0.05) and 4% lower perceived geographical barriers to care (b=−0.39, p<0.05). However, the associations were not statistically significant after adjustment for subnational facility and population characteristics (table 3). Coverage of family planning services was also negatively associated with the provision of outreach services. However, this paradoxical association was no longer significant in the multivariable specification. Meanwhile, no statistically significant association was found between any of the examined PHC quality subdomains and maternal health outputs (table 3).

On the other hand, PHC service delivery subdomains were more likely to be statistically significantly associated with child health outputs. For example, a 10%-point increase in the availability of basic facility infrastructure was associated with an increase of 5% in DPT3 immunisation coverage (b=0.45, p<0.05) and a decrease of 3% in DPT-dropout rates (b=−0.30, p<0.05). Additionally, recent facility supervision (b=0.34, p<0.01) and provision of feedback by supervisors (b=0.28, p<0.05) were each associated with increased care-seeking for pneumonia after adjustment for facility and population characteristics. Surprisingly, significantly lower immunisation coverage and higher dropout rates were observed in subnational areas where facilities did not charge PHC fees, even after adjustment for facility and population characteristics (table 4). Meanwhile, of the examined quality subdomains, only comprehensiveness showed a statistically significant association with subnational DPT3 immunisation coverage and lower dropout rates. Specifically, a 10%-point increase in the comprehensiveness of PHC services was associated with 2% improvement in DPT3 coverage and 2% lower DPT-dropout rates at the subnational level (b=0.22, p<0.05 and b=−0.21, p<0.01, respectively) (table 4).

**Table 3** The association between PHC service delivery subdomains and maternal health outputs at the subnational level

| Subdomain | | | Antenatal | | Family planning | | Cost barrier | | Geographical barrier | |
|---|---|---|---|---|---|---|---|---|---|---|
| | | | Bivariate | Multivariable | Bivariate | Multivariable | Bivariate | Multivariable | Bivariate | Multivariable |
| Capacity | Facility infrastructure | β | 0.23 | 0.11 | 0.29 | 0.16 | −0.35* | −0.23 | −0.39* | −0.16 |
| | | SE | 0.13 | 0.13 | 0.16 | 0.19 | 0.15 | 0.14 | 0.15 | 0.16 |
| | Supervision in the past 6 months | β | −0.09 | 0.02 | −0.12 | 0.01 | 0.05 | −0.02 | 0.00 | −0.14 |
| | | SE | 0.08 | 0.08 | 0.09 | 0.10 | 0.09 | 0.08 | 0.10 | 0.09 |
| | Supervisor feedback | β | −0.06 | 0.01 | −0.06 | 0.03 | −0.03 | −0.09 | −0.02 | −0.11 |
| | | SE | 0.08 | 0.07 | 0.09 | 0.09 | 0.09 | 0.07 | 0.09 | 0.08 |
| | Outreach services | β | −0.07 | 0.00 | −0.13* | −0.09 | 0.04 | −0.01 | 0.09 | 0.04 |
| | | SE | 0.05 | 0.04 | 0.06 | 0.06 | 0.06 | 0.05 | 0.06 | 0.05 |
| | Not charging PHC fees | β | 0.01 | 0.01 | −0.01 | 0.00 | −0.02 | 0.00 | 0.02 | 0.06 |
| | | SE | 0.05 | 0.04 | 0.06 | 0.06 | 0.06 | 0.05 | 0.06 | 0.05 |
| Quality | Waiting time <60 min | β | −0.01 | 0.01 | −0.11 | −0.10 | 0.09 | 0.05 | 0.11 | 0.07 |
| | | SE | 0.06 | 0.05 | 0.08 | 0.07 | 0.07 | 0.06 | 0.08 | 0.06 |
| | Provider competence | β | 0.07 | 0.00 | 0.04 | 0.01 | −0.28 | −0.10 | −0.26 | −0.10 |
| | | SE | 0.15 | 0.13 | 0.19 | 0.18 | 0.19 | 0.14 | 0.19 | 0.15 |
| | Provider availability | β | −0.04 | −0.08 | −0.16 | −0.15 | −0.11 | −0.13 | 0.02 | 0.03 |
| | | SE | 0.09 | 0.07 | 0.11 | 0.10 | 0.10 | 0.08 | 0.11 | 0.08 |
| | Person centredness | β | −0.01 | −0.04 | 0.01 | −0.02 | −0.06 | −0.01 | −0.02 | 0.02 |
| | | SE | 0.05 | 0.04 | 0.06 | 0.06 | 0.06 | 0.05 | 0.07 | 0.05 |
| | Safety | β | 0.09 | 0.06 | 0.01 | −0.03 | 0.20 | 0.13 | 0.11 | 0.10 |
| | | SE | 0.08 | 0.07 | 0.11 | 0.11 | 0.11 | 0.08 | 0.11 | 0.09 |
| | Comprehensiveness | β | 0.09 | 0.09 | −0.05 | −0.02 | −0.20 | −0.09 | −0.14 | −0.05 |
| | | SE | 0.10 | 0.08 | 0.11 | 0.11 | 0.11 | 0.08 | 0.12 | 0.09 |

In multivariable models, results were adjusted for proportion of private facilities, hospitals, urban households, population in the fifth economic quintile and women with secondary education or more at the subnational level.
Afghanistan was excluded from the analysis for subnational sample limitation.
*P value<0.05.
PHC, primary health care.

## DISCUSSION

Overall, the results of the descriptive analysis illustrate the heterogeneity in the capacity and quality of PHC service delivery within LMICs. The two-level analysis of maternal health outcomes yielded only a few statistically significant results at the p<0.05 level, however, none remained significant after adjusting for other covariates. This suggests that factors outside health facilities' control may play a greater role in determining mothers' care-seeking decisions for antenatal care and family planning, and in shaping perceptions of cost and geographical barriers to care. For example, individual-level and household-level factors such as the mother's education and socioeconomic status are associated with the use of family planning and antenatal care services. Nonetheless, supply-side constraints at the health system level such as supply chain challenges may also contribute to stockouts of essential family planning commodities.[18] Analysis at levels more granular than the subnational area (the first administrative level) may yield further insight into the relationship of supply-side PHC characteristics and maternal health outcomes.

Meanwhile, the two-level analysis of child health outcomes revealed significant associations between supervision and care-seeking for pneumonia, as well as associations of several measures of capacity and quality with DPT3 immunisation. Supervision in the last 6 months and supervisor feedback are considered measures of PHC capacity in the PHCPI framework, and it is hypothesised that measures of capacity operate through improved quality of health services. In the case of care-seeking for pneumonia, it is possible that facilities with greater capacity and implementation of supervision deliver care perceived to be of higher quality, thus increasing mothers' likelihood of seeking healthcare for their child at the facility. One review finds inconclusive but promising evidence that supportive supervision can improve quality of care, noting that supervision alone is insufficient; health workers require training and the availability of medicines, diagnostics and equipment to provide quality care.[19] Patient perceptions of a facility's quality of care also influence use and care-seeking.[20–26] There is also ample literature demonstrating the influence of various household and individual-level factors such as cultural beliefs, disease severity, geographical location, wealth, education and socioeconomic status on care-seeking behaviour.[27 28]

**Table 4** The association between PHC service delivery subdomains and child health outputs at the subnational level

| Subdomain | | | DPT3 | | DPT-dropout | | Pneumonia care seeking | | ORS coverage | |
|---|---|---|---|---|---|---|---|---|---|---|
| | | | Bivariate | Multivariable | Bivariate | Multivariable | Bivariate | Multivariable | Bivariate | Multivariable |
| Capacity | Facility infrastructure | β | 0.54* | 0.45† | −0.37* | −0.30† | 0.18 | 0.23 | −0.02 | 0.01 |
| | | SE | 0.15 | 0.18 | 0.12 | 0.14 | 0.18 | 0.22 | 0.12 | 0.15 |
| | Supervision in the past 6 months | β | −0.04 | 0.08 | 0.03 | −0.07 | 0.23† | 0.34* | 0.08 | 0.08 |
| | | SE | 0.09 | 0.10 | 0.07 | 0.08 | 0.11 | 0.12 | 0.07 | 0.08 |
| | Supervisor feedback | β | −0.03 | 0.05 | 0.01 | −0.05 | 0.22† | 0.28* | 0.10 | 0.09 |
| | | SE | 0.09 | 0.09 | 0.07 | 0.07 | 0.10 | 0.11 | 0.07 | 0.07 |
| | Outreach services | β | −0.07 | −0.02 | 0.06 | 0.01 | −0.04 | −0.05 | 0.05 | 0.04 |
| | | SE | 0.06 | 0.06 | 0.04 | 0.05 | 0.07 | 0.07 | 0.05 | 0.05 |
| | Not charging PHC fees | β | −0.14† | −0.17* | 0.07 | 0.09 | 0.09 | 0.09 | 0.08 | 0.07 |
| | | SE | 0.06 | 0.06 | 0.05 | 0.05 | 0.07 | 0.07 | 0.05 | 0.05 |
| Quality | Waiting time <60 min | β | −0.12 | −0.11 | 0.08 | 0.07 | 0.06 | 0.06 | −0.03 | −0.03 |
| | | SE | 0.07 | 0.07 | 0.06 | 0.05 | 0.08 | 0.08 | 0.06 | 0.06 |
| | Provider competence | β | 0.09 | 0.03 | −0.06 | −0.02 | 0.13 | 0.14 | −0.22 | −0.19 |
| | | SE | 0.19 | 0.18 | 0.14 | 0.14 | 0.21 | 0.21 | 0.14 | 0.14 |
| | Provider availability | β | −0.18 | −0.17 | 0.13 | 0.12 | 0.11 | 0.11 | −0.02 | −0.06 |
| | | SE | 0.10 | 0.10 | 0.08 | 0.08 | 0.12 | 0.12 | 0.08 | 0.08 |
| | Person centredness | β | −0.08 | −0.10 | 0.06 | 0.08 | 0.04 | 0.03 | −0.01 | −0.02 |
| | | SE | 0.06 | 0.06 | 0.05 | 0.05 | 0.07 | 0.07 | 0.05 | 0.05 |
| | Safety | β | 0.01 | 0.03 | −0.03 | −0.04 | 0.07 | 0.07 | −0.09 | −0.07 |
| | | SE | 0.11 | 0.11 | 0.08 | 0.08 | 0.12 | 0.13 | 0.09 | 0.09 |
| | Comprehensiveness | β | 0.23† | 0.22† | −0.22* | −0.21* | −0.10 | −0.11 | 0.03 | 0.03 |
| | | SE | 0.11 | 0.10 | 0.08 | 0.08 | 0.13 | 0.13 | 0.09 | 0.08 |

In multivariable models, results were adjusted for proportion of private facilities, hospitals, urban households, population in the fifth economic quintile and women with secondary education or more at the subnational level.
Afghanistan was excluded from the analysis for subnational sample limitation.
*P value<0.01.
†P value<0.05.
DPT, diphtheria-pertussis-tetanus; ORS, oral rehydration solutions; PHC, primary healthcare .

Finally, our results found several associations with DPT3 immunisation and dropout that generally align with existing literature (facility infrastructure and comprehensiveness) and in the case of user fees, counter to what might be expected. In line with our findings, prior research in Haiti found that both infrastructure quality and the quality-of-service delivery were associated with increased use of primary care services, including antenatal care and vaccination.[29] Similarly, a study in rural India found that children in districts where more villages had a healthcare subcentre were more likely to receive at least one dose of DPT3, though no association with DPT3 dropout rates was observed.[30] The results above also suggest that comprehensiveness, a measure of PHC quality, is also associated with better immunisation outcomes. This is in line with prior studies documenting that children whose mothers were counselled during vaccination sessions were more likely to complete all three DPT doses.[25]

One possible explanation behind the counterintuitive negative association between the absence of user fees and DPT3 immunisation is that the estimated association may be confounded by other factors determining both user fee exemptions and immunisation coverage.

For example, subnational areas that are generally poorer and have greater barriers to care may be more likely to adopt policies exempting user fees, yet those characteristics may also result in lower immunisation rates. A second explanation under the interpretation that the observed association is unconfounded may be related to potential unintended consequences of user fee removal. While the benefits of removing user fees are well documented in boosting health service usage,[31–33] some have also expressed concerns that the process of user fee removal must be designed carefully such that facilities are able to adapt. The elimination of fees and reduction of fees without a corresponding increase in resources from government sources could unintentionally lead to increased pressure on human resources and the availability of essential commodities, including vaccines. Further, increased demand for curative services without an increase in the health workforce could result in curative services 'crowding out' preventative services such as vaccination. Ultimately, further research into any potentially paradoxical relationship between user fees and DPT3 vaccination is needed.

The study should also be interpreted in light of its limitations. As the study is observational and cross-sectional, the association presented should not be interpreted as causal effects. Using data aggregated at the subnational level also entails the potential for an ecological fallacy and requires caution in interpretation. In addition, while the sample was constructed such that the time between the SPA and DHS surveys in each country was less than 5 years, it is possible that changes in time-varying factors or other events in that interlude could influence the outcomes under study. Afghanistan's more limited sample size restricted to predominantly urban areas presents an additional limitation, therefore it was excluded from regression analysis to mitigate any potential confounding. Finally, though we control for selected characteristics at the subnational and individual levels, there is a potential for omitted variable bias through other factors that may influence the capacity, quality and performance variables under study but are unmeasured in the SPA and DHS surveys. Ultimately, further research into the relationships between PHC capacity, quality and service delivery outcomes could adopt other methodologies to establish potential causal links. Taking advantage of repeated survey waves may provide one such strategy to track how these indicators may have changed over time and to what extent improvements in capacity and quality lead to better results.

Our findings have important policy implications. The heterogenicity observed in PHC service delivery within and between subnational areas highlights that there is not one solution that fits all. Countries and stakeholders seeking to strengthen their PHC systems could improve PHC monitoring at the subnational level. Disaggregation of national indicators by geographical regions and adoption of subnational scorecards can potentially unmask real bottlenecks that hinder the overall country's progress. Furthermore, the low average scores of provider competence and availability across the studied countries point to the need for investing in ongoing training and professional development of PHC providers. In addition, a participatory and mixed-methods approach to monitoring is important to better understand the diverse challenges and access barriers across subnational areas and develop tailored approaches to resolve the problems identified.

## CONCLUSION

Substantial heterogeneity and disparities were observed across the 12 studied LMICs, as well as between and within subnational areas in both the capacity and quality subdomains. Countries seeking to strengthen their PHC systems could improve PHC monitoring at the subnational level to better understand the diverse challenges across subnational areas and develop tailored approaches to resolve the problems identified. The associations observed between capacity, quality and performance also provide potential avenues for further research and

contribute to the literature on the determinants of care-seeking behaviour and immunisation.

**Contributors** MR and MV-U were responsible for the conception of this study and for methods development. MR analysed data. MR, JCG and CF prepared the first draft. All authors (MR, JCG, CF, SB, OB, MV-U) reviewed results or reviewed and contributed to the final manuscript. MR is responsible for the overall content as a guarantor.

**Funding** This research was supported, in whole or in part, by the Bill and Melinda Gates Foundation (Grant number INV000932), and it had no role in study design, data collection, data analysis, data interpretation or writing of the manuscript.

**Competing interests** None declared.

**Patient and public involvement** Patients and/or the public were not involved in the design, or conduct, or reporting, or dissemination plans of this research.

**Patient consent for publication** Not applicable.

**Ethics approval** Not applicable.

**Provenance and peer review** Not commissioned; externally peer reviewed.

**Data availability statement** The original data sets analysed during the current study are available from the DHS programme website, https://dhsprogram.com/data/available-datasets.cfm. The analytical data set generated during the current study is available from the corresponding author on reasonable request.

**ORCID iDs**
Marwa Ramadan http://orcid.org/0000-0003-4953-7346
Jose Carlos Gutierrez http://orcid.org/0000-0002-1489-2690
Cameron Feil http://orcid.org/0000-0002-9689-7240
Sarah Bolongaita http://orcid.org/0000-0003-0886-0454
Oscar Bernal http://orcid.org/0000-0003-2514-3318
Manuela Villar Uribe http://orcid.org/0000-0003-3386-4414

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
