## [Reviewer comments · BMJ Open]

ARTICLE DETAILS

TITLE (PROVISIONAL)	Capacity and quality of maternal and child health services delivery at the subnational primary health care level in relation to intermediate health outputs: a cross-sectional study of 12 low-income and middle-income countries
AUTHORS	Ramadan, Marwa; Gutierrez, Jose Carlos; Feil, Cameron; Bolongaita, Sarah; Bernal, Oscar; Villar-Uribe, Manuela

VERSION 1 – REVIEW

REVIEWER	Wang, Hong Bill & Melinda Gates Foundation
REVIEW RETURNED	05-Jul-2022

GENERAL COMMENTS	Thanks to the authors for their outstanding efforts to exam the relationships between inputs-process and outputs of PHC system, which is one of the very challenging issues to the researchers and policymakers. It is unsurprised that the associate relationships are weak. Here are a few suggestions for authors to consider. 1. Is it possible to create the summary index indicators of capacity, quality, and outputs and measure the association relationships among them, which could potentially increase/decrease the “powers” of the associations?2. Is it possible to show the effects of adjustment indicators on outputs and analysis which adjustments significantly affect the outputs?3. Some capacity indicators can also be considered quality indicators, as the authors rightly pointed out in the paper. Is it worth reclassifying some of these indicators into different categories and redoing the estimations?
--

REVIEWER	Azadnajafabad, Sina Tehran University of Medical Sciences
REVIEW RETURNED	07-Aug-2022

GENERAL COMMENTS	The authors of this study provided a multinational investigation on subnational heterogeneity in primary healthcare (PHC) services delivery in a group of low- and middle-income countries using the Demographic Health Survey (DHS) and Service Provision Assessment (SPA) datasets. The study design is robust and the manuscript is well prepared. Also, the results are promising and provide beneficial material for national and subnational health policymakers to resolve disparities in this regard. Prior to reaching a decision on this submission, some minor and major comments and suggestions need to be addressed and answered to improve the draft and resolve the concerns. Comments are listed below.
---

	1. Title, lines 1-2: since authors investigated only the PHC services delivery in the field of maternal and child health, it is highly suggested to reflect this issue in the title of study to prevent misleading and generalization of all PHC systems evaluation in this study. 2. Abstract, lines 19-43: the authors may provide the full-term of abbreviations in the first use in the abstract regardless of the main manuscript, since this section should stand alone and carry the whole message of study independently. 3. Abstract, lines 32-39: the results provided here need more numbers and rather than interpretations to convince readers about the achieved study results. 4. Introduction, lines 60, 61, 66, 69: the use of PHC abbreviation is poor and authors need to revise the text in this regard. 5. Introduction, lines 87-92: highlighting the disparities of healthcare services and PHC coverage at subnational level is a major part that needs to be addressed in this section since the main goal of this study is to cover this gap. Therefore, expanding the discussion on this very issue is essential. 6. Methods, lines 98-99: the public availability of the used data sources and the links to the used webpages to download the datasets for this study are suggested to be added to this part of the methods. 7. Methods, lines 100-102: one of the major concerns about this study is the reason for choosing these 12 countries. We know the authors indicated the low- and middle-income countries were the goal but why these countries were chosen? One possible reason could be the availability of the data on aimed health outcomes in these countries. This reason or any others that affected the country selection for study should be stated clearly in this section. Also, the index used to categorize the countries in this income level should be mentioned, which seems to be the World Bank Income groups. 8. Methods, lines 118-138: appropriate referencing to the PHCPI resources or publications is needed to make access to the metrics possible and make the methods stronger. 9. Methods, lines 153-162: more details on the used statistical methods, bivariate and multivariate models, the significance level of tests, and utilized softwares are needed in this section since authors did not provide such details. 10. Results, lines 167, 197: the main body of study results begins with national statistics and inter-country comparisons, while the reader expects to see results on subnational findings as it was highlighted in previous sections. I suppose the presentation of results are fine and the authors just need to mention they also provided between country investigations in the introduction and methods sections to not surprise readers in this section. 11. Results, line 227: "ANC" poor use of abbreviation. 12. Results, lines 241-242, 261-262: the details of the model need to be expanded in the methods section as comment number 9. Also, it seems authors recruited "multivariable" regression model rather than a "multivariate" model. These two models are different and authors need a revision and clarification about this concern. 13. Discussion, line 265: the authors indicate they provided a descriptive analysis in this line and some other parts of the study, while they recruited statistical models and the results include analytic parts, too. A revision in this regard may highlight the higher power of this investigation rather than a simple descriptive study.
--	--

	14. Discussion, lines 271-315: some paragraphs are suggested to be added in this section discussing the many interesting findings of this study using literature. Also, one especial paragraph highlighting the public health implications of the findings of this study and the beneficiaries for health managers and policymakers in resolving subnational health gaps and disparities in the investigated countries could upgrade this section. 15. References, lines 374-376: this citation needs revision in the name of organizations to properly project them. 16. Supplementary methods: the provided supplementary table 1 is a very informative table and I suggest authors move this table to the main text for a better presentation of primary data of the included countries in this study.
--	--

REVIEWER	Rahman, Aminur International Centre for Diarrhoeal Disease Research, Health Systems and Population Studies Division (HSPSD)
REVIEW RETURNED	09-Aug-2022

GENERAL COMMENTS	Need to mention clearly year of survey DHS and SPA data by country 12 countries facility distance from patient home is a major confounding variable how this was considered in this analysis and need to describe under each category which are the variables are included for universal understand Need to discuss detail operation definition of used quality, performance and outcome variables Why quality score of Bangladesh is absent Lets discuss a bit how quality score was measured as per WHO guideline You had chance to link with outcome variables by countries I think this is major limitation of this study. Need to mention recommendation of findings by country, These are very necessary for policy makers
---

REVIEWER	Marthias, Tiara The University of Melbourne, Nossal Institute for Global Health
REVIEW RETURNED	09-Aug-2022

GENERAL COMMENTS	This is an interesting paper that included a good range of LMICs and present an important topic on strengthening the primary health care system. However, the results seemed to be counterintuitive in several points, as has also been pointed out by the authors. Other than describing the limitations of the study and the datasets, I would suggest that the authors could explore other statistical approach, for instance multilevel analysis that could capture variations at different levels that may help explain better the variations in the outcome variables. Abstract  • Suggest clarifying that the outcome variables are on maternal and child health services. This was also not too clear in the title of the manuscript Introduction  • Page 4 line 34, missing a period. Methods
--

	 • Need more information on how SPA dataset matches the DHS sampling sites. Are the facilities included in the SPA sampled from the same clusters used in the DHS? • Page 5, line 119: spell out PHCPI • For family planning met need, more comprehensive definition should be clearly written and included in the Supplementary table as well. For instance, it was not clear whether the target population (women of reproductive age group) are ones who are on demand for FP services (wants to space childbirth, does not want anymore children, etc). • Would it be possible to use a multilevel modelling so that the analysis could control variables at various levels? E.g. individual, household, region/sub-national. This would also probably help explain better the variations in the outcomes variables. Discussion  • Taking into account that previous literatures have established that individual and household level factors may largely contribute to service coverage (and also supported by the lack of associations shown in this study), it might be better to explore other study design and/or statistical analysis and include these variables using the DHS datasets. • This would also encourage more specific policy inputs as to what aspects of the supply-side of the health system that are amenable and would help improve health care coverage, instead of just implying that the variations in service coverage may be better explained by variables not thoroughly included in this study, i.e. individual and household level.
--	--

VERSION 1 – AUTHOR RESPONSE

II. Response to Reviewer: 1

Dr. Hong Wang, Bill & Melinda Gates Foundation

Comment	Response / changes to the manuscript
1. Is it possible to create the summary index indicators of capacity, quality, and outputs and measure the association relationships among them, which could potentially increase/decrease the “powers” of the associations?	We would like to thank the reviewer for his suggestion. We calculated a summary index for capacity and quality at the subnational level as indicated in supplementary results [supplementary table 2 (Capacity) and supplementary table 3 (Quality)]. However, the heterogeneity observed within the same subnational area across individual capacity and/or quality indicators made us refrain from testing the association of indices with outputs (they may cancel each other and/ or they are mediators to one another). In addition, the research team was interested in understanding how each of the capacity and quality indicators is related to PHC output.

	We also refrained from summarizing output indicators as one index given the interest in understanding the association with each area of care e.g immunization, sick childcare, ANC, or family planning in addition, some of the tested outputs were difficult to aggregate as one index; some may function as mediators to the others e.g maternal access to care was an upstream indicator to ANC coverage and family planning coverage
2. Is it possible to show the effects of adjustment indicators on outputs and analysis which adjustments significantly affect the outputs?	We would like to thank the reviewer for the great suggestion. Yes, we did observe interesting effects for some adjustment factors e.g wealth quintiles, urbanicity of population as well as facility characteristics. Given that the analysis involved 12 LMICS and such interesting effects would need more elaboration. The plan was reporting initial descriptive results and having subsequent in-depth publications looking at inequalities/ diversities in PHC service delivery in the studied countries where we can have more flexibility to elaborate on adjustment factors and contrast results
3. Some capacity indicators can also be considered quality indicators, as the authors rightly pointed out in the paper. Is it worth reclassifying some of these indicators into different categories and redoing the estimations?	We agree with the reviewer that there are different ways for classifying the studied indicators. However, we would like to clarify that our methodology and proposed classification was based on PHCPI (Primary Healthcare Performance initiative) framework definitions for both capacity and quality. The framework consider that capacity indicators are upstream in nature to quality so capacity and quality are not mutually exclusive but rather capacity need to be in place (considered a mediator) for the delivery of quality services. Therefore, we echo the reviewer in the point raised. In future studies, we would be interested to examine/ test if the studied indicators would fit a different classification based on other health systems frameworks.

Reviewer: 2

Dr. Sina Azadnajafabad, Tehran University of Medical Sciences

Comment	Response / changes to the manuscript
1. Title, lines 1-2: since authors investigated only the PHC services delivery in the field of maternal and child health, it is highly suggested to reflect this issue in the title of study to prevent misleading and	We agree with the reviewer that PHC is more than maternal and child health. However, the indicators included in the present study constituted around 80 % of the indicators defined by the primary healthcare performance initiative

generalization of all PHC systems evaluation in this study.	(PHCPI) on which our methodology is based. Therefore, we kept the title as PHC. To avoid confusion, in line 29 of the updated manuscript [abstract] we indicated that the study output/outcome only involve eight maternal and child health outputs. We also expressed such limitation by stating that other PHC-related outputs (e.g., non-communicable diseases, tuberculosis and HIV) were not assessed due to the limited availability of sub-national data in the studied countries [lines 143-145]
2. Abstract, lines 19-43: the authors may provide the full-term of abbreviations in the first use in the abstract regardless of the main manuscript, since this section should stand alone and carry the whole message of study independently.	Corrections were made in the updated manuscript as advised by the reviewer [lines 21-45], also indicated by track changes in the marked copy
3. Abstract, lines 32-39: the results provided here need more numbers and rather than interpretations to convince readers about the achieved study results.	We added more numbers in the updated abstract [lines 37 and 38] as advised by the reviewer and within the maximal allowed word count
4. Introduction, lines 60, 61, 66, 69: the use of PHC abbreviation is poor and authors need to revise the text in this regard.	Corrections were made to lines 63, 64, 69, and 70 in the updated manuscript as suggested by the reviewer [also marked by track changes in the marked copy]
5. Introduction, lines 87-92: highlighting the disparities of healthcare services and PHC coverage at subnational level is a major part that needs to be addressed in this section since the main goal of this study is to cover this gap. Therefore, expanding the discussion on this very issue is essential.	Revisions were made to the paper as suggested by the reviewer [lines 87-89 in the updated manuscript]. Specifically, by adding the following: “However, the majority of associations were explored at the national level and significant gaps still exist in understanding PHC service delivery at the sub-national level and its association with improved health outputs. “
6. Methods, lines 98-99: the public availability of the used data sources and the links to the used webpages to download the datasets for this study are suggested to be added to this part of the methods.	The suggested revisions were added to data sources section [lines 118 and 119] including citation to the DHS website through the following statement “The SPA and DHS datasets utilized in this study are publicly available

	and can be downloaded from DHS website (15).“ Specific links are also provided as part of data availability statement [lines 358-360]
7. Methods, lines 100-102: one of the major concerns about this study is the reason for choosing these 12 countries. We know the authors indicated the low- and middle-income countries were the goal but why these countries were chosen? One possible reason could be the availability of the data on aimed health outcomes in these countries. This reason or any others that affected the country selection for study should be stated clearly in this section. Also, the index used to categorize the countries in this income level should be mentioned, which seems to be the World Bank Income groups.	The suggested changes were added to data sources section [lines 1001-103] by adding the following statement: “Specifically, we included all LMICS (as defined by the world bank income classification) where both a DHS and a SPA survey were conducted within a 5-year interval in the period between 2007 and 2019.”
8. Methods, lines 118-138: appropriate referencing to the PHCPI resources or publications is needed to make access to the metrics possible and make the methods stronger.	The requested clarifications were added in line 137 through the following statement: “Additional information on PHCPI’s framework and methodology is provided in an earlier publication (5).”
9. Methods, lines 153-162: more details on the used statistical methods, bivariate and multivariate models, the significance level of tests, and utilized softwares are needed in this section since authors did not provide such details.	The requested changes were made in lines 162-170 of the updated manuscript
10. Results, lines 167, 197: the main body of study results begins with national statistics and inter-country comparisons, while the reader expects to see results on subnational findings as it was highlighted in previous sections. I suppose the presentation of results are fine and the authors just need to mention they also provided between country investigations in the introduction and methods sections to not surprise readers in this section.	Revisions were made to the updated manuscript [lines 92-94] by adding the following statement “Specifically, we assess PHC service delivery at both the national and subnational levels in 12 LMICs. In addition, we examine the association of PHC capacity and quality with intermediate health outputs such as coverage and access to services at the subnational level.”
11. Results, line 227: “ANC” poor use of abbreviation.	Corrections were made throughout the manuscript as indicated by track changes in the marked copy
12. Results, lines 241-242, 261-262: the details of the model need to be expanded in the methods section as comment number 9. Also, it seems authors recruited “multivariable” regression model rather than a “multivariate” model. These two models are different and authors need a revision and clarification about this concern.	Revisions were made to the manuscript to clarify that a multivariable model was used throughout the paper as suggested by the reviewer [indicated by track changes]. Explanation of the analysis model was clarified as per comment number 9 [lines 162-170]

13. Discussion, line 265: the authors indicate they provided a descriptive analysis in this line and some other parts of the study, while they recruited statistical models and the results include analytic parts, too. A revision in this regard may highlight the higher power of this investigation rather than a simple descriptive study.	Revisions were made to the different parts of the manuscript to differentiate between the initial descriptive assessment of the heterogeneity of capacity and quality indicators and the two-level fixed effect modeling used for the assessment of sub-national associations [indicated by track changes]
14. Discussion, lines 271-315: some paragraphs are suggested to be added in this section discussing the many interesting findings of this study using literature. Also, one especial paragraph highlighting the public health implications of the findings of this study and the beneficiaries for health managers and policymakers in resolving subnational health gaps and disparities in the investigated countries could upgrade this section.	Updated paragraphs were added to the discussion section to clarify policy recommendations as advised by the reviewer [Lines 338-345]
15. References, lines 374-376: this citation needs revision in the name of organizations to properly project them.	Revisions were made to the introduction text for other requested clarifications, so the specified references are no longer used in the updated manuscript
16. Supplementary methods: the provided supplementary table 1 is a very informative table and I suggest authors move this table to the main text for a better presentation of primary data of the included countries in this study.	We moved the Table to the main body of the manuscript as suggested by the reviewer and renumbered tables accordingly

Reviewer: 3

Dr. Aminur Rahman, International Centre for Diarrhoeal Disease Research:

Comment	Response / changes to the manuscript
Need to mention clearly year of survey DHS and SPA data by country	We would like to thank the reviewer for his suggestion. We moved supplementary table 1 from supplementary methods to the article main body to indicate the year of both DHS and SPA survey for each country. .
12 countries facility distance from patient home is a major confounding variable how this was considered in this analysis and need to describe under each category which are the variables are included for universal understand	We agree with the reviewer that this is an important confounding factor to explore. In the present analysis, the linkage was done at the first administrative level for both the facility and population characteristics as explained in supplementary figure 1 [supplementary methods]. Such approach to analysis may carry both the risk for ecological bias and omitting variable bias [for some unadjusted confounding variables] as indicated in our strengths and limitation section. Given that there is a difficulty of geographically linking individual household

	record [for privacy reasons] with their facility record using the utilized data sources, it was not possible to control for the distance between a facility and patient home. However, other types of surveys/data sources may include questions of the specific geographic distance between the facility and the patient home, which may facilitate such analytic approach. In addition, another analytic approach (using DHS and SPA) may involve calculating a proxy using the centroid of DHS household cluster and the nearest facility, but the latter may still be subjected to misclassification error (assuming that the patient sought care at the nearest facility) Details on each of the variables utilized (under each category) and their specific definition are provided in supplementary table 2 [supplementary methods]
Need to discuss detail operation definition of used quality, performance and outcome variables	Details on each of the variables utilized (under each category) and their specific definition are provided in supplementary table 2 [supplementary methods]
Why quality score of Bangladesh is absent	The 2017 SPA datasets in Bangladesh were missing key variables for calculation of provider competence, patient centeredness and availability of providers as defined by PHCPI framework; however, we still included quality scores for each of the subnational areas in Bangladesh based on comprehensiveness and safety subdomains (as indicated in Table 1 [article main body] and supplementary table 3 in supplementary results)
Lets discuss a bit how quality score was measured as per WHO guideline	Details on PHCPI methodology for measuring quality base on WHO guidelines are included in earlier publication[as indicated in the metrics section], detailed definition of each of the utilized quality indicators/ tracer items are provided in supplementary Table 2 [supplementary methods]

VERSION 2 – REVIEW

REVIEWER	Azadnajibabad, Sina Tehran University of Medical Sciences
REVIEW RETURNED	13-Oct-2022
GENERAL COMMENTS	With many thanks, the authors addressed almost all of my comments and suggestions adequately and also provided reasonable answers to the raised queries of mine. I have no further comments on the revised manuscript.

REVIEWER	Marthias, Tiara The University of Melbourne, Nossal Institute for Global Health
REVIEW RETURNED	24-Oct-2022

GENERAL COMMENTS	Title & Abstract  • Grammar issue: One solution fits all. • As suggested in the first round of review, I believe it would be helpful to clearly state that the paper focuses on maternal and child health services at the primary care level. The current title implies that the paper covers a wide range of PHC services, while in reality, there are some more than just MNCH services. Strengths and limitations of this study  • Line 60, Typo: Middle-income □ middle-income • Also line 60-62: unclear what the research gap is • Line 63: I believe the findings showing how the variables included in the SPA are mostly not significantly associated with the outcome variables would imply otherwise, i.e. that the SPA survey did not manage to capture process indicators that are sensitive to the quality of care. Or can also be interpreted that the process indicators did not capture the quality of the activities that were measured (e.g. supervision may have occurred, but because it was not done properly, the quality of services delivered was not affected and remained low). • Line 72: “There is a potential for omitted variable bias” □ I believe omitted variable bias can only be established if there are one or more variables that were originally included in the regression but were then later omitted. In this paper’s case, the variable was never available. Methods  • Thank you for addressing previous comments on the variable definitions • And I appreciate the efforts made to exploring the random effect model and explaining why individual-level variables cannot be used as part of the multilevel analysis. • Line 368: “proportion of private facilities, proportion of hospitals...” □ proportion to what? And how was this calculated and at which level? This information is not available in the main text or the appendix. I think it’s great that the authors included these variables as it would capture some of the access issue, but also need to be careful with the different ratio/proportion used, e.g. proportion to population vs proportion to geographical area. The latter might be more sensitive for areas with less dense population and more remote areas. While the previous (ratio to population) could give misinterpretation where high ratio would be observed in less densely-populated areas but are usually also less developed. Results  • Line 532: “...and in the case of user fees, counter to what might be expected...” □ can the authors please elaborate on the possible explanation for this finding? • Line 534: “quality-of-service delivery were associated increased utilization ...” □ typo. Maybe were associated with increased utilization...
--

VERSION 2 – AUTHOR RESPONSE

II. Response to reviewer 4 comments

Title & Abstract  • Grammar issue: One solution fits all. • As suggested in the first round of review, I believe it would be helpful to clearly state that the paper focuses on maternal and child health services at the primary care level. The current title implies that the paper covers a wide range of PHC services, while in reality, there are some more than just MNCH services. [Note from the editors: this also applies to the abstract 'Objectives' section, which also should be more specific/precise 	Changes were made to the title as recommended by the reviewer and editor The current title is “Capacity and quality of maternal and child health services delivery at the subnational primary health care level in relation to intermediate health outputs: a cross-sectional study of 12 low-income and middle-income countries” The change was also reflected in the abstract as advised by the editor
Strengths and limitations of this study  • Line 60, Typo: Middle-income ◊ middle-income • Also line 60-62: unclear what the research gap is • Line 63: I believe the findings showing how the variables included in the SPA are mostly not significantly associated with the outcome variables would imply otherwise, i.e. that the SPA survey did not manage to capture process indicators that are sensitive to the quality of care. Or can also be interpreted that the process indicators did not capture the quality of the activities that were measured (e.g. supervision may have occurred, but because it was not done properly, the quality of services delivered was not affected and remained low). 	Line 60 corrected [Line 49 in the updated manuscript] Line 60-62 edited to demonstrate research gap [Lines 47-49 in the updated manuscript] Line 63: we agree with the reviewer that this is a possible explanation; however, we would like to clarify that we the journal recommendation that this section mainly refers to the strengths and the limitations of the methods of the present study regardless of the findings. Therefore, we recognized the importance of utilizing public available surveys as DHS and SPA in understanding bottlenecks in service delivery. In addition, when we look at the results from the SPA separately, we were able to identify several bottlenecks in the capacity and the quality of the system regardless of the association with intermediate health outputs. The latter may be affected by the methodological limitations of the present study [as we did not test for causality to ascertain that SPA survey did not capture the necessary process indicators], the inability to capture additional key tracer items or as explained by the reviewer, the tracer items captured through SPA survey

	did not manage to capture capacity or quality indicators that related to health outputs
 Line 72: "There is a potential for omitted variable bias" ◇ I believe omitted variable bias can only be established if there are one or more variables that were originally included in the regression but were then later omitted. In this paper's case, the variable was never available. 	We agree with the reviewer that this is one possible case of the omitted variable bias. In addition, we believe the bias can also happen if there is a potential confounding factor; but data was not available to account for it in the regression model, which is the case in the present analysis [as explained in the strengths and limitations section, lines 59-61 in the updated manuscript]

VERSION 3 – REVIEW

REVIEWER	Marthias, Tiara The University of Melbourne, Nossal Institute for Global Health
REVIEW RETURNED	30-Dec-2022

GENERAL COMMENTS	Thank you for addressing previous comments and making the necessary revisions. I have few inputs on grammar/sentence structure as follow: Line 339-340: Suggestion: "documenting that children of mothers"  children whose mothers were counseled or received counseling... Line 344-345, slight typo: "... subnational areas that are generally poorer and have greater barriers to care may be more likely adopt policies exempting user"  to adopt. Line 355-356: "Ultimately, further research into any potentially paradoxical relationship between user fees and DPT3 vaccination."  did the authors mean further research is needed? Line 377: "that hinder the overall country progress. " country's progress. care seeking  care-seeking. Ethics statement (line 390-391): "This study was based on secondary analysis of anonymized publicly available datasets. No ethics approval, or patient consent was need for the present analysis."  ... based on a secondary analysis.... And need to revise into: "was needed for the present". And finally, I believe the policy recommendations could be strengthened a bit further... for instance, what supports do PHCs need to ensure that comprehensive counseling for mothers are provided during vaccination sessions? What are the barriers based
--

	on literatures? Perhaps to also link this with the limited HR capacity (in the paragraph below the statement).
--	--

VERSION 3 – AUTHOR RESPONSE

Comment	Changes
Line 339-340: Suggestion: "documenting that children of mothers"  children whose mothers were counseled or received counseling...	Changed as requested -Line 306
Line 344-345, slight typo: "... subnational areas that are generally poorer and have greater barriers to care may be more likely adopt policies exempting user"  to adopt.	Changed as requested - Line 311
Line 355-356: "Ultimately, further research into any potentially paradoxical relationship between user fees and DPT3 vaccination."  did the authors mean further research is needed?	Changed as requested -Line 322
Line 377: "that hinder the overall country progress. " country's progress.	Changed as requested -Line 342
care seeking  care-seeking.	Changed as requested throughout the document
Ethics statement (line 390-391): "This study was based on secondary analysis of anonymized publicly available datasets. No ethics approval, or patient consent was need for the present analysis."  ... based on a secondary analysis....	Changed as requested - Line 357
And need to revise into: "was needed for the present".	Changed as requested -Line 358
And finally, I believe the policy recommendations could be strengthened a bit further... for instance, what supports do PHCs need to ensure that comprehensive counseling for mothers are provided during vaccination sessions? What are the barriers based on literatures? Perhaps to also link this with the limited HR capacity (in the paragraph below the statement).	Changes were implemented to the paragraph as advised by the reviewer 338-347

Response to Comments

On behalf of the authors, I would like to thank the editors and all the reviewers for their thoughtful comments to improve the quality of this manuscript. Below is a point-by-point response to reviewer 4 indicating any changes made to the manuscript